# Degradation of the UV Filter Benzophenone-4 by Ferrate (VI) in Aquatic Environments

Rouyi Wang [1], Ping Sun [2], Zhicai Zhai [2], Hui Liu [2], Ruirui Han [1], Hongxia Liu [1,*] and Yingsen Fang [2,*]

[1]  College of Advanced Materials Engineering, Jiaxing Nanhu University, Jiaxing 314001, China
[2]  College of Biological and Chemical Engineering, Jiaxing University, Jiaxing 314001, China
*   Correspondence: liuhongixa@zjxu.edu.cn (H.L.); fangyingsen@zjxu.edu.cn (Y.F.)

**Abstract:** This work demonstrates the potential utility of ferrate(VI)-based advanced oxidation processes for the degradation of a representative UV filter, BP-4. The operational parameters of oxidant dose and temperature were determined with kinetic experiments. In addition, the effects of water constituents including anions ($Cl^-$, $HCO_3^-$, $NO_3^-$, $SO_4^{2-}$), cations ($Na^+$, $K^+$, $Ca^{2+}$, $Mg^{2+}$, $Cu^{2+}$, $Fe^{3+}$), and humic acid (HA) were investigated. Results suggested that the removal rate of BP-4 (5 mg/L) could reach 95% in 60 min, when [Fe(VI)]:[BP-4] = 100:1, T = 25 °C and pH = 7.0, The presence of $K^+$, $Cu^{2+}$ and $Fe^{3+}$ could promote the removal of BP-4, but $Cl^-$, $SO_4^{2-}$, $NO_3^-$, HA and $Na^+$ could significantly inhibit the removal of BP-4. Furthermore, this Fe(VI) oxidation processes has good feasibility in real water samples. These results may provide useful information for the environmental elimination of benzophenone-type UV filters by Fe(VI).

**Keywords:** benzophenone-4; ferrate (VI); oxidation; kinetics

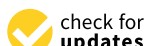



## 1. Introduction

Benzophenone (BP)-type UV filters are a kind of oil soluble substance that is easy to dissolve in aromatic hydrocarbons such as benzene and toluene but difficult to dissolve in water. Therefore, it is easier for this substance to enter the organism than water-soluble substances, thus causing toxic effects (Table 1 lists the structure and solubility of some substances). BPs are discharged into the sewage treatment system in a large amount and are difficult to degrade, and there are a large number of residues in the activated sludge. At the same time, BPs can also be brought into water through water entertainment activities such as swimming and bathing [1]. BPs have been detected in various water samples [2] and other environmental media [3,4]. Furthermore, BPs can also enter the human body through the skin, diet and air, which poses a threat to human health. BP residues have been detected in human samples [5,6]. Song et al. [7] reported that BP-3 concentrations showed positive correlations between maternal serum (MS) and cord serum (CS), while the CS/MS ratios of BP-1, BP-3, BP-8 and 4-OH-BP were affected by molecular weight or log$K_{ow}$, along with negative correlations.

**Table 1.** The structure and solubility of Part BPs.

| NO. | Name | Structure | Solubility |
|-----|------|-----------|------------|
| 1 | Benzophenone | | Insoluble in water (<0.1 g/100 mL at 25 °C) |

**Table 1.** *Cont.*

| NO. | Name | Structure | Solubility |
|---|---|---|---|
| 2 | Benzophenone-3(BP-3) |  | Insoluble in water (<0.1 g/100 mL at 20 °C) |
| 3 | Benzophenone-4(BP-4) |  | DMSO (Slightly), Methanol (Slightly) |
| 4 | 4-hydroxy benzophenone |  | Insoluble in water |

Passive diffusion may play an important role in the placental transfer of these BP-type UV filters. BP-3, BP-4, BP2 and 4-OH-BP were frequently detected in featured aquatic environments of Shanghai, and BP, BP-4 and BP-3 might adversely affect fish and other aquatic organisms [8].

Since BP substances are harmful to organisms and human, removing them from the environment is very important. Photochemical transformation is an important transformation pathway of UV sunscreen in natural water, which affects its environmental fate and ecological risk. The reaction rate and pathway of photochemical transformation are affected by pH and soluble substances in the water environment, and products with greater ecological risk can be produced under the light of BPs [9]. Vione et al. researched the light conversion process of BP-3 under the conditions related to surface water and found that BP-3 can be directly photodegradated and can also be indirectly photodegradated by hydroxyl radical (OH) and soluble organic matter DOM* [10]. The photolysis half-life of BPs varies with the environmental conditions and is possible in a few days to several months. Generally, organic pollutants can be removed by microbial degradation. Gago Ferrero et al. [11] studied the degradation and photodegradation processes of BP-1 and BP-3 by aerobic bacteria. The results showed that more than 99% of BP-1 and BP-3 could be degraded by biological treatment within 24 h, while the removal efficiency of phase reflective degradation was very low, especially for BP-3, which had little degradation effect. The results of metabolite analysis showed that BP-1 was decomposed by *T. versicolor* of BP-3, but the glycoconjugate derivative was the main metabolite. Moreover, there was no metabolite formation with a higher estrogen effect during the biodegradation of BP-1 and BP-3. Liu et al. [12] used activated sludge and digested sludge to study the biodegradation process under anaerobic and aerobic conditions. The results showed that anaerobic biodegradation was more suitable for the removal of BP-3. Beel et al. [13] explained the anaerobic biodegradation process of BP-4 in the activated sludge of the urban sewage treatment system and found nine kinds of transformation products that showed a higher toxicity of bacteria (*Vibrio fischeri*) than BP-4.

As this emerging contaminant cannot be eliminated effectively by conventional processes, it is necessary to explore efficient measures for the degradation of BPs from water. In order to remove UV filters from water, many measures have been researched, including ozonation, chlorine, persulfate and ferrate [14,15]. Amongst them, the products of ferrate ($FeO_4^{2-}$, Fe(VI)) reaction are ferric oxides/hydroxides, which can act as coagulants/precipitants. Therefore, as an effective green oxidant, ferrate(VI) (Fe(VI)) has attracted much attention [16,17]. Research works have proved that Fe(VI) is very promising for the

degradation of organic materials, such as PPCPs, endocrine disrupting chemicals and pesticides [18–21]. However, information on the degradation of BP-4 by Fe(VI) is scarce.

In this study, we attempted to investigate the oxidation of BP-4 by Fe(VI) in aquatic environment. First, experiments were implemented in order to obtain the optimal reaction conditions of oxidant dose and temperature. Then, coexisting water components such as metal cations ($Na^+$, $K^+$, $Ca^{2+}$, $Mg^{2+}$, $Cu^{2+}$, $Fe^{3+}$), inorganic anions ($Cl^-$, $NO_3^-$, $HCO_3^-$, $SO_4^{2-}$) and humic acid (HA) were tested for their effects on BP-4 removal. Finally, the removal of BP-4 in natural waters was also evaluated.

## 2. Materials and Methods

### 2.1. Chemicals and Reagents

BP-4 (CAS no: 4065-45-6, 98% purity) and potassium ferrate ($K_2FeO_4$, CAS no: 39469-86-8 Fe(VI), purity > 95%) were obtained from J&K Company (Shanghai, China). Methanol and formic acid gained from Merck Company (Darmstadt, Germany) (for HPLC) were of HPLC grade. The rest of the reagents were of analytical grade or higher. Ultrapure water (18.2 MU cm) was prepared with a Milli-Q system (Millipore, Bedford, MA, USA).

### 2.2. Removal of BP-4 by Fe(VI)

Degradation experiments were carried out in batches with 100 mL brown glass in a rocking bed (150 r/min) at $25.0 \pm 0.2$ °C. The pH of solution was initially adjusted by HCl or NaOH. At the specified time interval, an 0.8 mL sample was collected and immediately filtered into a 2.0 mL vial containing 0.2 mL chromatographic grade methanol to quench the reaction (through a 0.22 μm filter). In order to study the potential effects of various environmental factors, the initial BP-4 solution was adjusted to a different pH value or pre-added with 0.5 mM ions ($Fe^{3+}$, $Ca^{2+}$, $Mg^{2+}$, $Cu^{2+}$, $Na^+$, $K^+$, $SO_4^{2-}$, $Cl^-$, $NO_3^-$, $HCO_3^-$) and 1–30 mg/L humic acid (HA). The degradation of BP-4 in two environmental water samples in Jiaxing (one surface water sample from canal, one sample of effluent of Jiaxing domestic sewage treatment plant in Jiaxing, China) was also studied. All experiments were carried out twice, and the average values are presented in the paper.

### 2.3. Analytical Methods

The concentrations of BP-4 were measured by an Agilent 1200 high performance liquid chromatograph (HPLC) equipped with a quaternary pump and a diode array detector (Agilent Technologies, Palo Alto, CA, USA). Chromatographic analysis was performed with a 1.0 mL/min flow rate on a Zorbax Eclipse XDB-C18 analytical column (4.6 mm × 150 mm, particle size 5 mm) (Agilent Technologies, CA, USA) at 30 °C. The injection volume was 20 μL, and the elution time was 10 min for all samples. The mobile phase was methanol (A) and water (B) (60:40).

## 3. Results

### 3.1. Removal of BP-4 from Aqueous Solution

Initially, removal of BP-4 by Fe(VI) was investigated with different molar ratios of Fe(VI) to BP-4. The results are presented in Figure 1. The concentration of BP-4 decreased rapidly within 10 min and changed slightly after 60 min. This is because the strong oxidation of Fe(VI) can effectively remove the target pollutants in a very short time when the oxidant has a high initial concentration in the reaction solution. As time goes by, the concentration of Fe(VI) decreases rapidly, and the reaction tends to stop. In addition, the removal rate of BP-4 increases remarkably with increasing Fe(VI) (except [Fe(VI)]:[BP-4]) = 200:1). For example, the removal rate BP-4 was from 30% to 95% (at 60 min) when the molar ratio was from 25: 1 to 100: 1. At a higher ratio of 200:1, the degradation efficiency will be lower. This is because Fe(VI) reacting with the target pollutant is enough in the reaction solution and will only overflow as the dosage increases. In addition, the instability of ferrate will lead to the auto-degradation of Fe(VI), and the auto-degradation rate will also increase with the increase of the concentration of Fe(VI) solution. The ratio of Fe(VI) used to degrade BP-4

decreases, meaning that the degradation efficiency decreases. Feng et al. also reported this phenomenon in their research [20].

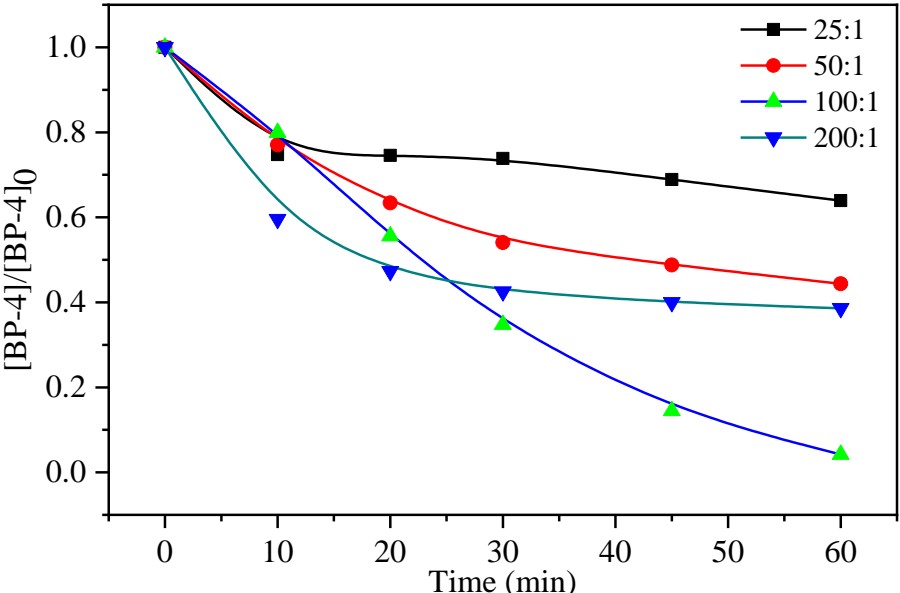

**Figure 1.** Effect of dose of oxidants ([Fe(VI)) on the removal of BP-4 (experimental conditions: $[BP-4]_0$ = 5 mg/L; pH = 7.0 ± 0.2; T = 25.0 °C; reaction time = 90 min).

In the next set of experiments, the effects of pH on the degradation were determined, as shown in Figure 2A. The removal rate of BP-4 was greatly affected by the pH value of the solution. The reason is that the pH value of the solution will affect the oxidation ability and stability of potassium ferrate. In acid conditions, potassium ferrate has the highest oxidation potential (2.2 ev) in conventional water treatment agent, which is much higher than 0.72 ev in alkaline condition, meaning the degradation rate of pH = 3 (highest) is much higher than that of pH = 11 (lowest) in the first 10 min. There are different forms of ferrate solution in different pH environments—generally, the forms are $H_3FeO_4^+$, $H_2FeO_4$ and $HFeO_4^-$. In the acidic environment, the ferrate in the aqueous solution is mainly in the form of $H_2FeO_4$ and $HFeO_4^-$, at which time the ferrate is active and easy to decompose; in the alkaline condition, the ferrate is mainly in the form of stable $FeO_4^{2-}$. Therefore, the stability of potassium ferrate is better when the pH is higher than 7. When the pH is 3, the ferrate root is extremely unstable, and its self-decomposition is intensified, which leads to basic non-degradation after 10 min, and the final degradation rate is the lowest. Potassium ferrate with pH = 7 and pH = 9 has both high oxidation potential and good stability, meaning these should be the best pH values for degradation of BP-4. At pH = 11, its good stability enables ferrate to react with BP-4 continuously. Thus, the next experiment was studied with pH = 7.0.

The reaction temperature also significantly influenced the degradation of BP-4. As shown in Figure 2B, the BP-4 degradation rate is faster with the temperature increasing. The reason is that heating can accelerate the efficient degradation of BP-4 by Fe(VI), and the results are in accordance with the degradation of organic chemicals by other oxidants, such as potassium permanganate and persulfate [22,23]. The degradation rate constants (k) were 0.0204, 0.0290 and 0.0407 $min^{-1}$ at 25 °C, 35 °C and 45 °C. In addition, based on the experimental data, the activation energy (Ea) of the BP-4 oxidative degradation by Fe(VI) was estimated to be 27.2 KJ/mol ($R^2$ = 0.9999) with the Arrhenius equation. It can be speculated that higher temperatures were of benefit to the removal of BP-4.

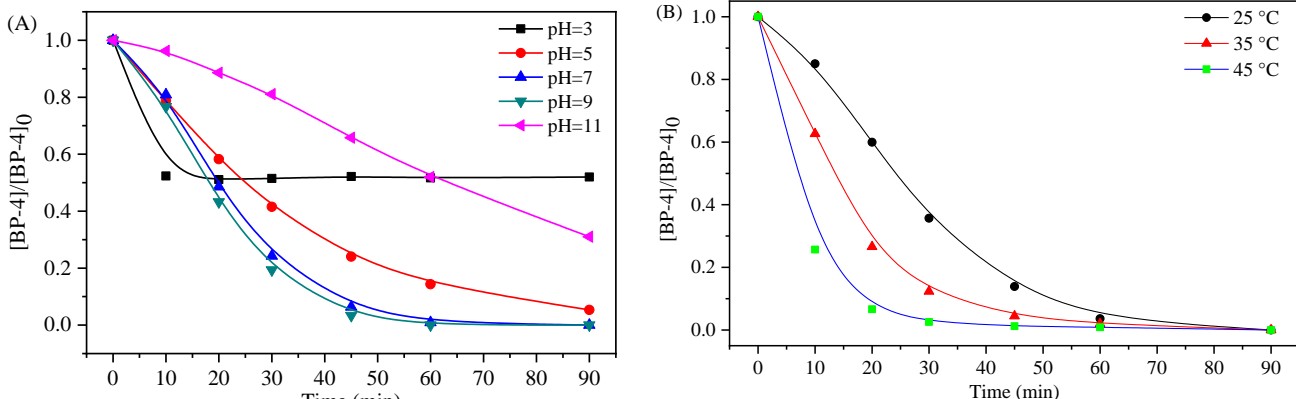

**Figure 2.** Effect of initial solution pH (**A**) and temperature (**B**) on BP-4 removal by Fe(VI) (experimental conditions: $[BP\text{-}4]_0$ = 5 mg/L; T = 25.0 °C for pH (initial pH 7.0 ± 0.2 for temperature); $[Fe(VI)]_0$:$[BP\text{-}4]_0$ = 100:1; reaction time = 90 min).

Based on the above results, experimental conditions were the following: [BP-4] = 5 mg/L, initial pH 7.0 ± 0.2, $[Fe(VI)]_0$:$[BP\text{-}4]_0$ = 100:1; T = 25 °C, reaction time = 90 min.

### 3.2. Effect of Inorganic Ions on Degradation

The effect of inorganic ions on the degradation of BP-4 was evaluated (as seen in Figure 3). Experimental conditions were as follows: [BP-4] = 5 mg/L, $[Fe(VI)]_0$:$[BP\text{-}4]_0$ = 100:1, T = 25 °C, [Anions] = [Cations] = 5 mM. As shown in Figure 3A, $Cl^-$, $SO_4^{2-}$ and $NO_3^-$ ions significantly inhibited the removal efficiency of BP-4 by Fe(VI), but the impact on BP-4 removal was not apparent with the addition of $HCO_3^-$ at 5 mM. For monovalent cations such as $Na^+$ and $K^+$, they had an obvious influence on BP-4 removal (Figure 3B). $Na^+$ ions reduced the removal of BP-4 by Fe(VI), while $K^+$ ions could promote its degradation. When $Mg^{2+}$ and $Ca^{2+}$ were added in reaction solution, the effects were not obvious. $Ca^{2+}$ ions inhibited the removal of BP-4 by Fe(VI), and $Mg^{2+}$ ions presented no obvious effect on BP-4 removal. Moreover, the effect of $Cu^{2+}$ and $Fe^{3+}$ (transition metal ions) on the removal of BP-4 was also investigated. When these ions were present in the reaction solution, the removal rate of BP-4 increased (Figure 3B). $Cu^{2+}$ showed more increase than $Fe^{3+}$.

According to Figure 3, $K^+$, $Cu^{2+}$ and $Fe^{3+}$ could promote the removal of BP-4, but $Cl^-$, $SO_4^{2-}$, $NO_3^-$ and $Na^+$ could significantly inhibit the removal of BP-4. The specific reason was that $K^+$, $Cu^{2+}$ and $Fe^{3+}$ play a catalytic role in the oxidative degradation of BP-4 by Fe(VI).

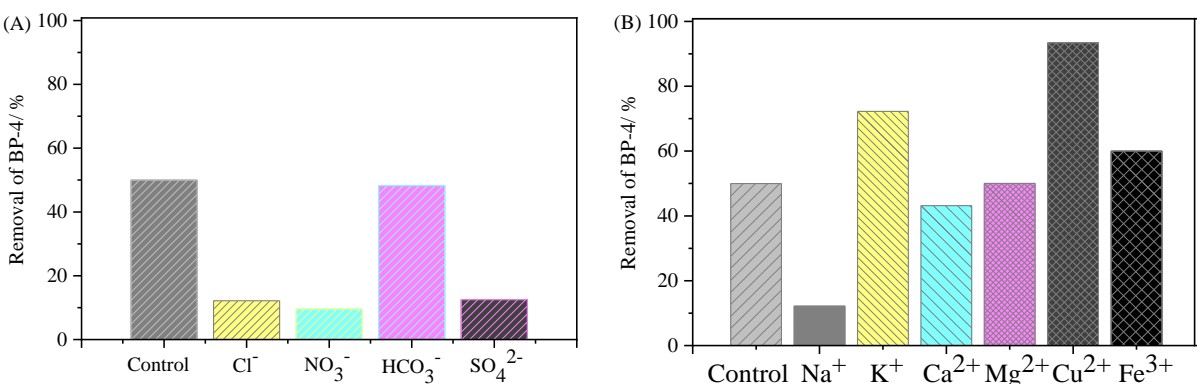

**Figure 3.** Effect of anions (**A**) and cations (**B**) on the removal of BP-4 after 10 min. [Anions] = [Cations] = 5 mM.

### 3.3. Effect of HA on Degradation

The effect of dissolved organic compounds on the removal of BP-4 by Fe(VI) was measured by adding HA with a concentration of 1–30 mg/L. The addition of HA decreased

the efficiency of removing BP-4 by Fe(VI) (Figure 4). With the increase of HA concentration, the removal rate of BP-4 nonlinearly decreased. The results show that HA competes with BP-4 to react with Fe(VI). In other words, Fe(VI) reacts not only with BP-4 but also with HA.

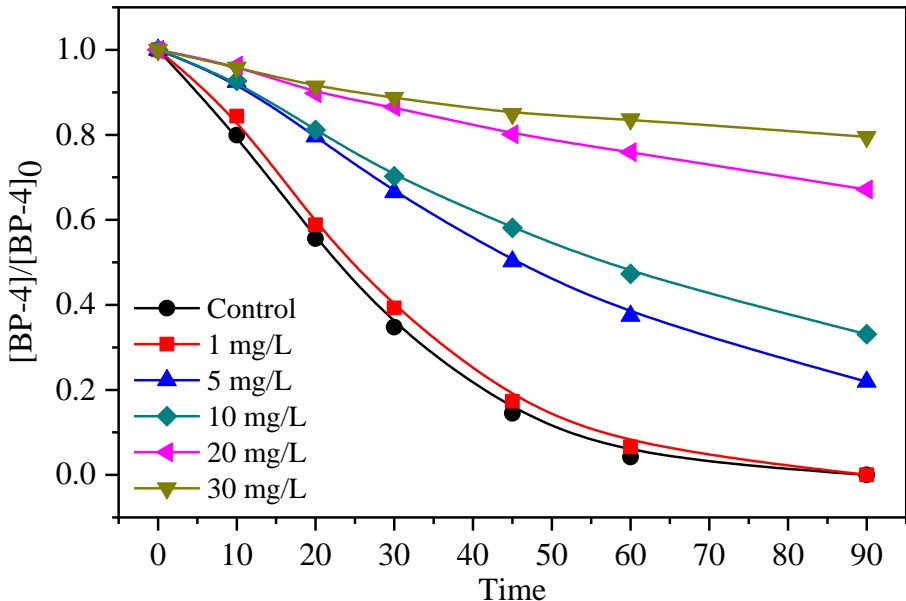

**Figure 4.** Effect of humic acid (HA) concentration on the removal of BP-4.

In Figure 4, when the HA concentration was lower than 1 mg/L, the removal rate of BP-4 did not show an obvious effect due to HA. The removal rate of BP-4 decreased with the increase of concentration of HA. When the concentration of HA increased to 20 mg/L, the removal rate of BP-4 changed from 95.82% to 24.09% in 60 min, and when the concentration of HA reached 30 mg/L, the removal rate of BP-4 was reduced to 16.31%. The main reason was that HA has a variety of functional groups, such as -COOH, -OH, etc. The organic matter with these functional groups could have different chemical reactions with oxidants in water, including ferrates. The HA could be oxidized by $FeO_4^{2-}$ in the reaction solution, and that would reduce its reaction with BP-4. At the same time, the HA could not have been oxidized completely, and its products could also form complexes or be subject to adsorption with the final product $Fe(OH)_3$ colloid of potassium ferrate, thus competing with BP-4 and reducing the removal rate of BP-4.

### 3.4. Removal of BP-4 in Environmental Water Samples

It is necessary to assess the feasibility of this oxidative technique to completely eliminate low levels of sunscreen agents in different waters. Initially, the removal of BP-4 by Fe(VI) in different waters was evaluated at pH 7.0. The molar ratio of Fe(VI) to BP-4 was 100:1. The results of the removal of BP-4 as a function of time from various water matrices are shown in Figure 5. Fe(VI) can almost completely remove BP-4 from most water samples in 90 min, except for secondary effluent water samples. The reason may be the coexisting constituents, which could significantly inhibit the removal of BP-4.

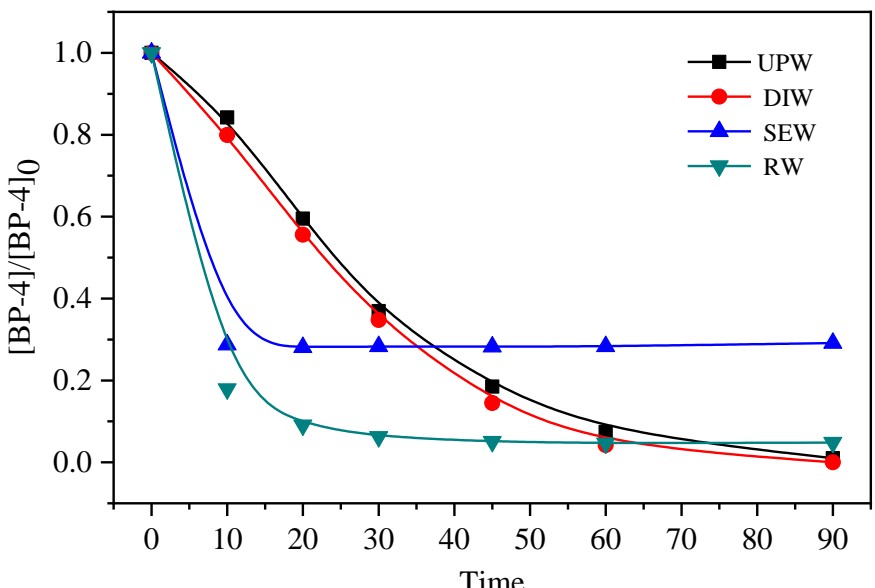

**Figure 5.** Removal of BP-4 from water samples by Fe(VI) at a molar ratio of 100:1 ($[Fe(VI)]_0$:$[BP-4]_0$). (abbreviations: UPW—ultrapure water; DIW—deionized water; SEW–secondary effluent of sewage treatment plant; RW—Jiaxing Canal River Water.).

### 3.5. Oxidation Products of BP-4 and Possible Reaction Pathways

To identify the identification of oxidation products, mass analysis experiments were performed at 5 mg/L BP-4 (initial concentration), [Fe(VI)]:[ BP-4] = 100:1, T = 25 °C and pH = 7.0. A total of three products were identified in positive mode by LC-TOF-MS, and structural assignments were achieved with the product ion scan. The MS/MS spectra and the proposed fragmentation patterns of BP-4 and its reaction intermediates are illustrated in Figure 6.

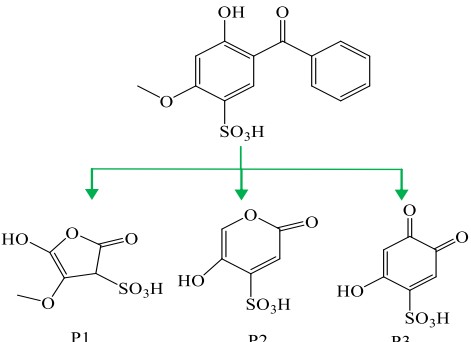

**Figure 6.** Degradation pathways of BP-4 by ferrate (VI) process.

Figure 6 shows that the ring of benzene can be oxidized, and the results are in accordance with references with PMS and ozone [14,24]. While the ring of benzene is not oxidized with the biological method [12], the ability of the chemical oxidizing agent is stronger than biodegradation, so more effective methods need to be developed.

## 4. Conclusions

The research suggests that Fe(VI) has great potential to remove BP-4 and can be used as a rapid and effective method. The presence of inorganic ions (i.e., $K^+$, $Cu^{2+}$ and $Fe^{3+}$) in water increased the removal efficiency of BP-4 by Fe(VI). In contrast, $Cl^-$, $NO_3^-$, $SO_4^{2-}$, $Na^+$, $Mg^{2+}$ and $Ca^{2+}$ ions decreased the removal efficiency of BP-4 by Fe(VI). HA may affect

the removal efficiency of BP-4 by Fe(VI). The removal of BP-4 by Fe(VI) varied with the water type and constituents of water matrices.

**Author Contributions:** H.L. (Hongxia Liu): conceptualization, validation, investigation, data curation, writing—original draft preparation, writing—review and editing, visualization, supervision, project administration, funding acquisition; Y.F.: conceptualization, validation, investigation, data curation, writing—original draft preparation, writing—review and editing, visualization, supervision, project administration, funding acquisition; R.W.: formal analysis, data curation, writing—original draft preparation, writing—review and editing; P.S.: formal analysis, investigation, writing—review and editing; Z.Z.: formal analysis, data curation, writing—review and editing; H.L. (Hui Liu): methodology, formal analysis, writing—review and editing; R.H.: formal analysis, writing—review and editing. All authors have read and agreed to the published version of the manuscript.

**Funding:** This work was funded by the National Natural Science Foundation of China—Project-ID 21607058; the Zhejiang Provincial Natural Science Foundation of China—Project-ID LY21B070008 and LY21B070007; the Department of Education of Zhejiang Province—Project-ID Y201840526; the Scientific Research Startup Foundation for Leading Professor from Jiaxing University—Project-ID CD70519063; the Student Research Innovation Team Funding Project of Zhejiang—Project-ID 2019R417027; the Public Welfare Research Project of Jiaxing—Project-ID 2020AY10005, 2021AY10069 and 2021AD10008; and National innovation and entrepreneurship training program for College Students—Project-ID 202113291013.

**Institutional Review Board Statement:** Not applicable.

**Informed Consent Statement:** Not applicable.

**Data Availability Statement:** All the data are available within the manuscript.

**Conflicts of Interest:** The authors declare no conflict of interest.

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
