# Peer review of "Degradation of the UV Filter Benzophenone-4 by Ferrate (VI) in Aquatic Environments"

_processes, doi:10.3390/pr10091829_

Round 1

Reviewer 1 Report

The manuscript is interesting but it needs extensive language corrections.  It contains several spelling errors (missing spaces) and, unfortunately, many language errors that seriously decrease the paper's quality.  In some sections there are instructions for the Authors using the Journal's template that should be deleated.                     

I would also recommend adding some more information on BP's physico-chemical properties (with structures, perheps), to explain the differences e.g. between BZ-3 and BZ-4 concerned with their ability to cross the skin barrier or to bioconcentrate in aquatic organisms  - it is important in the context of these compounds' toxicology.

Author Response

Dear Reviewer

I have received your review about the galley corrections for processes-1891799. As for questions referred, all responses presented point by point, are as follows:

  1. Extensive editing of English language and style required

Response: Thank you for useful and ponderable advice on our manuscript, and the manuscript has been extensively revised to make thorough and lucid. Furthermore, the amendments are modified format display in the revised manuscript. Hope this explanation will be more legible.

  1.  

Yes

Can be improved

Must be improved

Not applicable

Does the introduction provide sufficient background and include all relevant references?

(x)

( )

( )

( )

Are all the cited references relevant to the research?

(x)

( )

( )

( )

Is the research design appropriate?

(x)

( )

( )

( )

Are the methods adequately described?

(x)

( )

( )

( )

Are the results clearly presented?

( )

( )

(x)

( )

Are the conclusions supported by the results?

( )

(x)

( )

( )

Response: 

Thanks for your nice suggestion. Image has been replaced(see revised manuscript). And the conclusions were revised.

Comments and Suggestions for Authors

  1. The manuscript is interesting but it needs extensive language corrections. It contains several spelling errors (missing spaces) and, unfortunately, many language errors that seriously decrease the paper's quality. In some sections there are instructions for the Authors using the Journal's template that should be deleated.      

Response: Thank you for useful and ponderable advice on our manuscript, and the manuscript has been extensively revised to make thorough and lucid. Furthermore, the amendments are modified format display in the revised manuscript. Hope this explanation will be more legible.

  1. I would also recommend adding some more information on BP's physico-chemical properties (with structures, perheps), to explain the differences e.g. between BZ-3 and BZ-4 concerned with their ability to cross the skin barrier or to bioconcentrate in aquatic organisms - it is important in the context of these compounds' toxicology.

Response: Thanks for your nice suggestion. BP's physico-chemical properties with part structures were added in revised manuscript(see Table 1). But relevant data of their ability to cross the skin barrier or to bioconcentrate in aquatic organisms are not found, and will be further studied in the future.

We have also examined the manuscript thoroughly and make more accurate modification. We hope you and the reviewers could accept our rebuttals and provide us more comments so that we can improve our future work.

Thank you for your kind help and attention. I am looking forward to your good news.

Yours sincerely,

Hongxia Liu

Reviewer 2 Report

The authors did interesting work. However, some minor improvement is needed before the article acceptance

1. To highlight the present work, the comparison table is needed

2. Image quality need to be improved

3.  References are very old. Try to cite the recent research article.

4. Some spelling errors and grammatical mistakes need to be removed

Author Response

Dear Reviewer

I have received your review about the galley corrections for processes-1891799. As for questions referred, all responses presented point by point, are as follows:

  1. English language and style are fine/minor spell check required.

Response: Thank you for useful and ponderable advice on our manuscript, and the manuscript has been extensively revised to make thorough and lucid. Furthermore, the amendments are modified format display in the revised manuscript. Hope this explanation will be more legible.

  1. To highlight the present work, the comparison table is needed.

Response: Thanks for your nice suggestion. The comparison were added in revised manuscript. Figure 6 shows that the ring of benzene can be oxidized, the results are in accordance with references with PMS and ozone[14, 24]. While the ring of benzene isn’t oxidized with biological method[12]. The ability of chemical oxidizing agent is stronger than biodegradation, so more effective methods need to be developed.

  1. Image quality need to be improved.

 Response: Thanks for your nice suggestion. It has been replaced(see revised manuscript).

  1. References are very old. Try to cite the recent research article.

 Response: Thanks for your nice suggestion. It has been replaced such as references 1 and 2.

[1] Gago-Ferrero, P.; Mastroianni, N.; Diaz-Cruz, M.S.; Barcelo, D. Fully automated determination of nine ultraviolet filters and transformation products in natural waters and wastewaters by on-line solid phase extraction-liquid chromatography-tandem mass spectrometry, J. Chromatogr. A 2013, 1294, 106-116.

[1] Sánchez-Quiles, D., Blasco, J., Tovar-Sánchez, A. Sunscreen Components Are a New Environmental Concern in Coastal Waters: an overview. sunscreens coast. ecosyst. occur. behav. Eff. Risk 2020. 1-14. 

[2] Liu, Y.S.; Ying, G.G.; Shareef A.; Kookana, R.S. Occurrence removal of benzotriazoles and ultraviolet filters in a municipal wastewater treatment. Environ. Pollut. 2012, 165, 225-232. 

[2] Mao, F.; He, Y.; Gin, K.Y.H. Occurrence and fate of benzophenone-type UV filters in aquatic environments: a review. Environ. Sci. Water Res. Technol. 2019, 5, 209–223. 

We have also examined the manuscript thoroughly and make more accurate modification. We hope you and the reviewers could accept our rebuttals and provide us more comments so that we can improve our future work.

Thank you for your kind help and attention. I am looking forward to your good news.

Yours sincerely,

Hongxia Liu
